# Adult Intestinal Toxemia Botulism

**DOI:** 10.3390/toxins12020081

**Published:** 2020-01-24

**Authors:** Richard A. Harris, Fabrizio Anniballi, John W. Austin

**Affiliations:** 1Botulism Reference Service for Canada, Microbiology Research Division, Bureau of Microbial Hazards, Food Directorate, Health Products and Food Branch, Ottawa, ON K1A 0K9, Canada; Richard.Harris@canada.ca; 2National Reference Centre for Botulism, Microbiological Foodborne Hazard Unit, Department of Food Safety, Nutrition and Veterinary Public Health, Istituto Superiore di Sanità, viale Regina Elena, 29900161 Rome, Italy; fabrizio.anniballi@iss.it

**Keywords:** *Clostridium botulinum*, *Clostridium butyricum*, *Clostridium baratii*, botulism, botulinum toxin, intestinal toxemia

## Abstract

Intoxication with botulinum neurotoxin can occur through various routes. Foodborne botulism results after consumption of food in which botulinum neurotoxin-producing clostridia (i.e., *Clostridium botulinum* or strains of *Clostridium butyricum* type E or *Clostridium baratii* type F) have replicated and produced botulinum neurotoxin. Infection of a wound with *C. botulinum* and in situ production of botulinum neurotoxin leads to wound botulism. Colonization of the intestine by neurotoxigenic clostridia, with consequent production of botulinum toxin in the intestine, leads to intestinal toxemia botulism. When this occurs in an infant, it is referred to as infant botulism, whereas in adults or children over 1 year of age, it is intestinal colonization botulism. Predisposing factors for intestinal colonization in children or adults include previous bowel or gastric surgery, anatomical bowel abnormalities, Crohn’s disease, inflammatory bowel disease, antimicrobial therapy, or foodborne botulism. Intestinal colonization botulism is confirmed by detection of botulinum toxin in serum and/or stool, or isolation of neurotoxigenic clostridia from the stool, without finding a toxic food. Shedding of neurotoxigenic clostridia in the stool may occur for a period of several weeks. Adult intestinal botulism occurs as isolated cases, and may go undiagnosed, contributing to the low reported incidence of this rare disease.

## 1. Introduction

Botulism is a rare but severe neuroparalytic disease caused by exposure to botulinum neurotoxins (BoNTs). Botulism can be classified, depending upon the route of exposure, into six forms. The disease may be naturally caused by ingestion of pre-formed toxin in food (foodborne botulism), infection of a wound with *Clostridium botulinum* resulting in toxin production in situ (wound botulism), colonization of the infant intestinal tract (infant botulism), and colonization of the intestinal tract of adults or children over 1 year of age (intestinal toxemia botulism). In addition to these naturally occurring forms of botulism, iatrogenic and inhalation botulism have been recognized. They are respectively due to the erroneous administration of toxin for therapeutic/cosmetic purposes, and by inhalation of accidental/deliberately aerosolized toxin [1]. 

The World Health Organization has reported an estimated 475 cases of foodborne botulism occur in Canada, Europe, and the United States each year. These cases result in prolonged physical disability in the majority of cases and lethality in 15% of cases [2]. Symptoms of botulism generally begin with cranial nerve palsies, resulting in one or more of ptosis, diplopia, fixed and dilated pupils, dysphonia, and dysphagia, followed by a descending symmetrical flaccid paralysis. A complete clinical review of the symptoms of botulism in adult patients was recently reported [3]. Globally, the most common form of botulism is foodborne botulism; however, in some countries such as the United States, infant botulism is the most common form of botulism with more than 100 cases recognized annually [4]. Symptoms of infant botulism include generalized weakness and hypotonia, lethargy, constipation, difficulty feeding, and cranial nerve palsies [5]. Typically, and historically, *Clostridium botulinum* has been recognized as the cause of botulism, however, neurotoxigenic strains of *Clostridium baratii* type F and *Clostridium butyricum* type E have also been recognized as causative agents of botulism. 

BoNTs are the most lethal poisons known [6]. On the basis of neutralization with specific antisera, they are classified into seven serotypes A through G. The serotypes are further divided into subtypes on the basis of amino acid sequences of the proteins [7]. In addition to serotypes A through G, BoNT/H (also known as F/A or H/A) has been described [8,9,10]. The availability of extensive bacterial DNA sequences has recently allowed discovery of botulinum toxin-like genes and corresponding toxins in a variety of bacteria, including non-clostridial species [11,12,13,14,15,16,17,18,19]. Recently, the first botulinum toxin targeting an invertebrate has been discovered [20]. BoNTs exert their action through their metalloprotease activity on SNARE (soluble N-ethylmaleimide-sensitive factor attachment protein receptors) proteins responsible for docking a fusion of small synaptic vesicles with the cytoplasmic face of the neuron plasma membrane [21,22,23,24]. Cleavage of SNARE proteins prevents the release of acetylcholine at the neuromuscular junction, resulting in flaccid paralysis. 

Adult intestinal toxemia botulism has been aptly described as “an elusive disease to classify” [25]. It shares an etiology with infant botulism. Both infant botulism and adult intestinal toxemia botulism are intestinal toxemias [26], or toxicoinfections [27], with BoNT-producing clostridia colonizing the intestinal tract and producing botulinum toxin in situ. The distinction between adult intestinal toxemia botulism and infant botulism is based on the age of the patient, and this form of botulism has been referred to as “infant botulism in adults” [28]. The disease is referred to by several names, all of which indicate colonization of the intestine or toxemia caused by neurotoxigenic clostridia. Keeping in mind the fact that the disease may occur in anyone over 1 year of age, the disease will be referred to as adult intestinal toxemia botulism in this review. 

This review describes adult intestinal toxemia botulism, highlighting its peculiarities with respect to the other forms of botulism, as well as highlighting the difficulties in recognition of cases which, in turn, may result in underestimation of the incidence of the disease.

## 2. Ecology of BoNT-Producing Clostridia in the Environment and in Foods

BoNT-producing clostridia are Gram-positive obligate anaerobes that are widely present in soils and sediments throughout land and coastal waters across the globe [29]. Contamination of foods with *C. botulinum* can occur during harvesting, processing, or storage, after which favorable environments could lead to growth and production of toxins [30]. An important feature of these bacteria is the ability to form endospores that are resistant to inactivation treatments common in food processing, including heat, high hydrostatic pressure, and ionizing radiation. Considering that clostridial spores are generally not harmful to healthy adults, and that harsh treatments may negatively affect food quality and increase production costs, it is not always practical to inactivate spores during food production.

The incidence of spores in foods depends largely on the distribution and incidence of neurotoxigenic clostridia in the environment [29]. A wide range of foods associated with foodborne botulism outbreaks have been traced to contamination with *C. botulinum* spores, including meats, fish, and vegetables [29]. For example, half of the samples of carrot juice sampled during an outbreak of foodborne botulism tested positive for *C. botulinum* [31], and close to three quarters of samples of vacuum-packed bacon tested positive in another study [32]. Although spores may be found in foods, the adult intestinal tract does not generally support spore germination and toxin production under normal circumstances [1]. Thus, it is not uncommon for food to contain spores of *C. botulinum*, and ingestion of spores occurs naturally without causing foodborne botulism. 

The source of spores is rarely determined in adult intestinal toxemia botulism cases. This is similar to infant botulism where, with the exception of honey [26], the source of the spores usually remains undiscovered. Foods have been determined as possible sources of spores for five cases. These foods include a refrigerated jar of cream of coconut [33,34], blackberries [34], tuna and/or spaghetti with meat sauce [35], vacuum-packaged hashed beef [36], and peanut butter [37]. Only in the case of peanut butter were strains from food and stool confirmed as genetically related by DNA typing. In addition to foods as a source of spores, it has been suggested that patients of infant botulism may inhale spores carried by dust that then stick to saliva and are swallowed [26]. The same may be true in the case of adult intestinal toxemia botulism.

## 3. Predisposing Factors

Although adult intestinal toxemia botulism has been reported in previously healthy individuals with no history of gastrointestinal problems [34,36,37,38], the majority of cases involve patients with previous bowel surgery, bowel anomalies, or recent use of antimicrobials that may disrupt the normal intestinal flora. Previous bowel and gastric surgery [33,34,37,39,40,41,42,43,44] appear to be predisposing factors, as does Crohn’s disease [37,44]. Previous administration of antibiotics, perhaps causing perturbation of the regular gut flora, has been reported for several cases [35,36,45,46]. Some cases have received immunosuppressive drugs, including prednisone [44] and cyclosporine [47]. 

Two cases of adult toxemia botulism were reported in Italy, with both cases having Meckel’s diverticulum, [48]. Meckel’s diverticulum is a congenital abnormality of the small intestine resulting from incomplete absorption of the vitelline duct during gestation. To date, these have been the only two cases colonized by *C. butyricum* type E. Additionally, diverticulitis was reported for a *C. baratii* type F case of adult intestinal toxemia botulism in the United States [49]. 

Two cases of adult intestinal toxemia botulism in allogeneic hematopoietic cell transplant recipients have been reported [45,47]. Alteration of the intestinal flora may occur as a result of antibiotics before and during hematopoietic cell transplant, administration of chemotherapy and/or radiation, and altered nutritional patterns [50]. One case has been recorded where the immunosuppressant cyclosporine was administered in the 53 days prior to symptom onset [47].

## 4. The Possible Role of the Gut Microbiome in Colonization

Intestinal toxemia botulism occurs when spores of BoNT-producing clostridia are ingested, transit to the intestinal tract, germinate, multiply, and colonize the lumen of the large intestine [26]. In the case of adult intestinal toxemia botulism, there may be a focus of infection, or the entire lumen may be colonized [44]. Although most cases of adult intestinal toxemia botulism have been caused by *C. botulinum*, *C. baratii* type F and *C. butyricum* type E have also been incriminated in certain cases [43,46,48]. 

Susceptibility to intestinal colonization by *C. botulinum* spores, but not by *C. baratii* type F and by *C. butyricum* type E, has been demonstrated through animal models. Adult rodents treated with antibiotics showed an increased susceptibility to colonization from ingested *C. botulinum* spores when compared to untreated controls, yet gained resistance to colonization when housed for just 3 days in a room with untreated/control adults [51,52]. Infant rodents were found to be naturally susceptible to colonization without the use of antibiotics, and required a much lower infective dose of *C. botulinum* spores for colonization compared with antibiotic-treated adult mice [53]. A peak susceptibility of infant rodents to botulinum colonization was demonstrated to be 7–13 days of age, which mirrored the age distribution of 2–4 months for infant botulism cases in the United States [51,53]. 

The intestinal tract of adults does not generally support spore germination and toxin production under normal circumstances [1]. Thus, it is not uncommon for food to contain spores of *C. botulinum*, and ingestion of spores, without botulinum toxin, occurs naturally without causing illness. A healthy gut microbiome is important for prevention of both neurotoxin absorption and *C. botulinum* colonization. In vitro experiments have suggested that probiotic bacteria can improve tight junction signaling in the intestinal epithelium [10] and decrease internalization of neurotoxins [3]. Although it is well established that the adult gut shows an increased resistance to colonization of *C. botulinum* spores, the exact mechanism of competitive microbial resistance remains to be determined. Several species of the established adult intestinal microflora, including those of the genus *Lactobacillus* and *Pediococci*, produce lactic acid and acetic acid that lower the pH and are known to prevent growth of *C. botulinum* [54]. Another potential mechanism may be mediated by active antagonism through the release of bacteriocins targeting related bacterial species. Experimental studies have shown beneficial effects of a bacteriocin produced by probiotic strains of *C. butyricum* in preventing infection in other *Clostridia*, including *Clostridium difficile* [55] and *Clostridium pasteurianum* [56], yet its effect on *C. botulinum* remains to be investigated. Additional mechanisms for competitive microbial resistance may include nutrient limitation, enhancing the intestinal mucosal barrier, and activation of host immunity [57].

## 5. Pathophysiology

BoNTs are primarily absorbed in the small intestine [58,59], where they are either translocated directly through intestinal epithelial cells or bypass a disrupted epithelial barrier [60,61]. It was hypothesized that BoNTs produced initially in the cecum might initially paralyze the ileocecal valve allowing the reflux of toxins into the small intestine where the absorption is more efficient [62], probably explaining why clinical features of intestinal toxemia botulism often show persistent neurotoxin in stool, but not always in serum. Absorption of BoNTs across the intestinal barrier occurs by receptor mediated transcytosis, in a temperature-dependent and saturable manner [60,61,63,64]. After crossing the intestinal epithelial barrier, BoNTs disperse in extracellular fluids, enter the lymphatic system, and then enter into the blood circulation. From general circulation BoNTs reach the target cells inducing neuronal intoxication via a four-step mechanism consisting of (i) high affinity and specific binding to receptors on the presynaptic membrane of skeletal and autonomic cholinergic nerve terminals, (ii) internalization by endocytosis, (iii) low-pH driven membrane translocation, and (iv) proteolysis of SNARE proteins [22,65]. This complex mechanism of action is functionally related to the structural organization of BoNTs that consists of three main domains, as reviewed elsewhere [24,65]. Proteolysis of the SNARE proteins prevents fusion of small synaptic vesicles with the cytoplasmic face of the plasma membrane, and subsequent release of acetylcholine.

## 6. Diagnosis and Treatment

Severity of adult intestinal toxemia botulism may range from relatively mild cases with hospitalization for a month [38], to severe cases with prolonged hospitalization and rehabilitation [36,66], to fatality resulting from complications of botulism [39]. The clinical presentation of adult intestinal toxemia botulism is similar to other forms of botulism. Initial symptoms may include one or more of nausea [46,47], vomiting [46,47,48], generalized muscle weakness [67], bloating [39] or distended abdomen [37,38,44,47,48,66], decreased bowel sounds [37,38], abdominal pain [39,46,47,48], and constipation [38,39,41,46,48,67], in some cases lasting weeks to months [36]. Cranial nerve palsies may result in one or more cases of blurred vision, diplopia, ptosis, ophthalmoplegia, dysarthria, and dysphagia. These symptoms occur in an otherwise alert and afebrile patient. The disease may progress to a descending, symmetrical flaccid paralysis, eventually causing respiratory failure requiring ventilation support [36,39,43,46]. Paralysis may progress to a point where the patient appears comatose [48,66]. Alternative diagnoses may include Miller Fischer variant of Guillain–Barré syndrome [37,39,68,69], myasthenia gravis [37,39,44,68], viral encephalitis [37], meningitis [37], Eaton–Lambert syndrome [37], brain stem lesion [37], brain death [66], appendicitis [48], or *Clostridium difficile* associated diarrhea [47]. 

Although examples of incidents where the same patients have experienced foodborne botulism more than a single time have been reported [70,71,72], suggesting that previous exposure does not illicit an immune response, and giving rise to the saying that “the toxic dose is less than the immunogenic dose”, evidence exists that patients with adult toxemia botulism may mount an immune response to botulinum toxin [44]. This may explain why botulinum toxin can initially be detected in patients’ sera but does not persist even though the colon remains colonized with *C. botulinum* [44].

Distinguishing between foodborne botulism and adult intestinal toxemia botulism is a diagnostic challenge, yet essential for the treatment of the patient and identification of potentially contaminated foods. The task depends upon communication between clinicians; epidemiologists; and provincial, state, or national reference laboratories, as detection of BoNTs or BoNT-producing clostridia is not performed by hospital laboratories. Until recently, adult intestinal toxemia botulism was recognized only after investigations ruled out other potential causes of disease [25]. However, recent cases have identified several key observations can help identify adult intestinal toxemia botulism. These include the prolonged excretion (10 days to over 100 days) of viable *Clostridium* spores or neurotoxin in the stool [34,36,37,43,48], persistent reoccurrence of disease symptoms [48], and a high level of botulinum neurotoxin in the feces concomitant with a low level of toxin in serum [26,37,43], although serum neurotoxin levels can fluctuate [48]. Further careful observations from future case studies will help to inform epidemiological investigations with the aim of detecting potential food sources of contamination from neurotoxin (foodborne botulism) or spores (intestinal toxemia botulism). 

Treatment of adult intestinal toxemia botulism is limited to administration of botulinum antiserum and supportive care, including mechanical ventilation if required. Patients may remain dependent upon a ventilator for several months [35,36,37]. Because there have been so few cases, there is no standard treatment. Antitoxin is not always administered [41,67]. Antibiotics are sometimes administered [36,39,41]; however, the usefulness of antibiotics has been questioned [28] as they cause further disruption of the normal intestinal microbiota [47] and spores may continue to be shed for weeks to months [37]. Promoting the return of a healthy gut microbiome in infants and adults may be an overlooked yet viable treatment option with the use of probiotics. Clostridia represent a large fraction of commensal microbiota of the healthy gut [73,74]. Non-toxigenic strains of *C. butyricum* have been used as probiotics to modulate the intestinal ecosystem [75] and suppress experimental colitis in mice [74]. Several different probiotic strains have also shown promising potential for directly blocking the epithelial internalization of botulinum neurotoxin type A in vitro, including *Saccharomyces boulardii*, *Lactobacillus acidophilus*, *Lactobacillus rhamnosus* LGG, and *Lactobacillus reuteri* [76]. Recently, the probiotic strain of *Lactobacillus reuteri* was used after antitoxin administration at a therapeutic dose of 1 × 10^9^ bacteria daily for a case of infant botulism [77]. Further investigation is warranted in animal studies to determine the potential therapeutic role of probiotics in restoring a healthy gut microflora after intestinal toxemia botulism. 

## 7. Recognition of Cases 

Adult intestinal toxemia botulism is exceptionally rarely reported, with just over 20 cases reported in the literature (Table 1); however, it may occur more commonly than is recognized [37,44]. Recognition of cases requires botulism be included in the initial differential diagnosis. A complete food history should be obtained and food specimens submitted for testing in order to determine whether a case is foodborne botulism. Finally, multiple stool specimens should be submitted and tested for botulinum neurotoxin as well as viable neurotoxigenic clostridia. 

Adult intestinal toxemia botulism does not occur as outbreaks, only as sporadic cases. Clinical recognition of foodborne botulism is more likely to occur when multiple cases with similar symptoms are observed, especially when multiple cases are members of the same family, have shared the same food, or report to the same hospital [78,79]. It is not uncommon for sporadic cases, and even cases in foodborne botulism outbreaks, to be misdiagnosed [31,79], often as Guillain–Barré syndrome or myasthenia gravis. Recognition of sporadic cases of a rare disease such as adult intestinal toxemia botulism requires a high index of suspicion by clinicians.

When clinical cases of botulism are recognized, and clinical samples submitted for analysis are found positive, a toxic food is often not found. Some of these may be adult intestinal colonization cases, and some cases may be foodborne but are not associated with a toxic food because foods were not submitted for testing. From 1976 to 2014, the U.S. Centers for Disease Control reported 73 cases of botulism in patients >1 year of age, that were not “definitively classified as either foodborne or wound” [25]. The Italian National Reference Centre for Botulism identified a food vehicle in 30.7% of all laboratory confirmed incidents (86/285) from 1986 to 2015 [80], whereas the Botulism Reference Service for Canada determined a toxic food in 78.3% of laboratory confirmed botulism cases from 1985 to 2005 [72]. Determination of a toxic food during an investigation depends upon follow-up at the regional public health level by obtaining a food history and submission of food specimens for analysis. The extent of follow-up at the local public health level differs depending upon regional health authorities. Outbreaks tend to elicit a more thorough investigation, whereas single cases (such as adult intestinal colonization cases) may not receive the same attention.

Additionally, adult intestinal toxemia botulism cases are difficult to confirm. A clinical case of botulism in a patient aged greater than or equal to 1 year, who has no history of ingestion of a suspect food and who has no wounds, should heighten suspicion of a possible case of intestinal toxemia botulism. If available, stool samples should be periodically taken and tested for the presence of viable neurotoxigenic clostridia. Receipt of stool samples over an extended period requires an engaged physician and arrangements with hospital staff. Botulinum toxin may be detected in serum or stool shortly after onset of symptoms, however, shedding of viable *C. botulinum* may occur for several weeks (Table 1). Although molecular typing of *C. botulinum* by whole genome sequencing and multilocus sequence typing (MLST) is now commonplace, molecular epidemiology is not necessary to implicate a food for foodborne botulism, as the finding of a toxic food finishes the epidemiological investigation. Unlike foodborne botulism, a toxic food is not available for epidemiological investigation of intestinal toxemia botulism—either infant or adult. Given the relatively high incidence of spores of *C. botulinum* in certain foods, molecular typing could be useful to link clinical and food isolates in the case of intestinal colonization botulism. Just as molecular typing of *Salmonella*, *Listeria*, or verotoxigenic *Escherichia coli* isolates from food and clinical samples is necessary to link a contaminated food to cases of illness.

In the absence of a suspected food source, it remains important to characterize isolated strains from cases of infant and adult intestinal toxemia botulism in order to inform future epidemiological studies and lead to better prevention strategies. For example, the vast majority of colonization botulism cases are a result of group I *C. botulinum*, *C. baratii*, and *C. butyricum* strains, presumably because they prefer a higher incubation temperature to group II *C. botulinum* strains [29]. Yet, recently a case of infant colonization botulism was reported as caused by a group II *C. botulinum*, type E, which have an optimal growth temperature of 25 °C [81]. Considering that adult intestinal toxemia botulism is difficult to diagnose and exceptionally rare, a concerted effort by healthcare professionals should be taken to identify and characterize this form of botulism. Whole genome sequencing of isolated clostridia would allow a detailed comparison of strains that are more likely to colonize the intestine from those associated with foodborne or wound botulism. This effort may reveal trends in distribution of spores that are prevalent in the environment and better predict their occurrence in contamination of food during production. Genomic characterization may also reveal novel colonization factors, or other mechanisms, that could be exploited to prevent growth in the gut and lead to more effective therapeutic strategies for colonization botulism.

## 8. Summary

Infant botulism is now well-recognised as a disease, but intestinal toxemia botulism in older children and adults is less well known and potentially underreported. The potential patient outcomes of adult intestinal toxemia botulism can be severe and may include prolonged debilitation and death. Patient prognosis is dependent upon rapid and accurate diagnosis based on clinical symptoms with laboratory confirmation by finding extended shedding of *C. botulinum* in stool specimens. A diagnosis of adult intestinal toxemia botulism should be considered in a patient displaying both gastrointestinal symptoms and cranial nerve palsies, particularly in those displaying the form of descending, symmetrical flaccid paralysis typical of botulinum intoxication. Particular consideration should be given to patients with a history of previous bowel surgery, Crohn’s disease, and anatomical abnormalities of the bowel or recent antimicrobial therapy. If suspected, intestinal toxemia botulism can be confirmed by detection of botulinum toxin in serum and/or stool, or isolation of neurotoxigenic clostridia from the stool, with the lack of an identifiable foodborne source.

## Figures and Tables

**Table 1 toxins-12-00081-t001:** Listing of adult intestinal toxemia botulism cases reported in the literature.

Year Published	Year of Case	Age/Sex	Symptoms	Possible Redisposing Factors	Length of Colonization ^1^	Length of Hospitalization	Toxin Type	Samples Testing Positive	Reference
**1980**	1980	47/M	Abdominal bloating, nausea, vomiting, diarrhea, diplopia, dry mouth, dysphagia, unreactive pupils, hypotension, ptosis, bilateral peripheral facial weakness, respiratory paralysis, colonic dilatation, partial ileus	None	N.D. ^2^	21 days	B	*Clostridium botulinum* type B in stool	[68]
**1981 and 1988**	1978	33/F	Dizziness, “thick tongue”, weakness, respiratory arrest	Jejunoileal bypass	N.D.	Died of respirator complications 17 days after admission	A	*C. botulinum* type A in stool; type A toxin in serum	[34,40]
**1986**	N.S.	45/F	Generalized weakness, fatigue, loose stools, gastric distension, constipation, dysphagia, diplopia, ptosis, facial weakness, bilateral tinnitus, slurred speech	Jejunoileal bypass	N.D.	22 days	B	*C. botulinum* type B in stool	[41]
**1988**	1973	Elderly/M	N.S. ^3^	N.S.	32 days	N.S.	B	Type B toxin in stool and serum; *C. botulinum* type B in stool	[34]
**1988**	1981	27/M	“classic signs of botulism”	Classical foodborne botulism outbreak with secondary cases	47 days	N.S.	B	Type B in serum; *C. botulinum* type B isolated from stool	[34]
**1986 and 1988**	1985	37/F	Weakness, dysarthria, diplopia, malaise, abdominal pain, constipation, dysarthria, dysphagia, otherwise alert and oriented, followed by dyspnea, ophthalmoplegia, bulbar paralysis, descending paralysis	Billroth I surgical procedure to remove pyloric valve, 5 weeks prior to admission for botulism	119 days	Died of polymicrobial sepsis after 240 days	A	Type A toxin in serum; type A toxin in stool; *C. botulinum* type A in stool	[33,34]
**1991**	1987	54/M	Dysarthria, diplopia, dysphagia, weakness, respiratory arrest, remained alert	Truncal vagotomy, pyloroplasty, and cholecystectomy 4 years prior	14 days	31 days; limited physical endurance 3 years later	F	*Clostridium baratii* type F in stool; *C. botulinum* type B in colonic washing	[43]
**1994**	1992	3/F	Slurred speech, progressive weakness, respiratory failure, constipation	Immunosuppression and antimicrobials pre- and post-autologous bone marrow transplant	N.D.	Died after 150 days	A and B	Types A and B toxins in stool	[45]
**1997**	1988	67/M	Abdominal pain, diplopia, decreased bowel sounds, abdominal distension, dysarthria, dyspnea, bilateral facial weakness, respiratory arrest	Crohn’s disease; terminal ileum and right colon had each been resected; treated with prednisone	19 weeks	79 weeks	A	*C. botulinum* type A in stool and gastric aspirate; type A toxin in serum, stool, and gastric aspirate	[44]
**1999 and 2007**	1994	9/M	Diplopia, bilateral mydriasis, dysphonia, dry mouth, dry eyes, constipation, tympanic abdomen, urinary retention, tachycardia, tachypnea, dyspnea, facial nerve palsies, upper limb weakness, respiratory failure, electromyography (EMG) compatible with botulism	Meckel’s diverticulum	16 days	37 days	E	*Clostridium butyricum* type E in stool	[48] [46]
**1999 and 2007**	1995	19/F	Abdominal pain, nausea, vomiting, diplopia, dysphagia, dry mouth, dysphonia, facial nerve dysfuntion, constipation, ptosis, mydriasis, respiratory failure, asthenia, coma, complete paralysis, EMG compatible with botulism	Meckel’s diverticulum	11 days	23 days	E	*C. butyricum* type E from rectal swab	[48] [46]
**2007**	1997	56/M	Diplopia, dysphagia, nausea, vomiting, afebrile, respiratory failure	Ceftriazone therapy	45 days	90 days	A	*C. botulinum* type A in stool; type A toxin in serum	[46]
**2002**	2001	41/F	Shortness of breath, weakness, vomited several times, dizziness, bradycardia, respiratory arrest, unreactive pupils	Amoxicillin for 2 days before symptom onset	N.D.	>12 weeks	F	Type F toxin in serum; *C. baratii* type F from spaghetti sauce and can of tuna ^4^	[35]
**2003**	1999	12/F	Constipation	Foodborne botulism, antibiotics	>122 days	425 days	Ab	*C. botulinum* Ab in stool; type Ab toxin in serum	[36]
**2012**	2006	63/F	Abdominal pain, blurred vision, diarrhea, dysarthria, dysphagia, diplopia, imbalance, weakness in arms and hands, opthalmoplegia, ptosis, respiratory arrest	Crohn’s disease, previous bowel surgery, short bowel syndrome	61 days	7 months	A	*C. botulinum* type A in enema fluid, stool, and gastric aspirate	[37]
**2012**	2007	50/F	Opthalmoparesis, dysphagia, quadriparesis	Crohn’s disease, four previous bowel resections, ileocolonic anastomosis, enterocutaneous fistulas	56 days	128 days in hospital; 4 months in rehabilitation hospital	B	Botulinum toxin in serum and stool; *C. botulinum* in stool	[37]
**2008 and 2012**	2008	45/M	Small bowel obstruction, distended abdomen, minimal bowel sounds, blurry vision, pupils unreactive to light	None identified	41 days	51 days	B	*C. botulinum* type B in stool	[37,38]
**2014**	2007	79/M	Diplopia, weakness, diarrhea, nausea, vomiting	Endoscopy 3 days prior to symptom onset	N.D.	Died on day 15 of acute respiratory failure resulting from paralysis and an underlying medical condition	F	Botulinum toxin in serum; type F toxin and *C. baratii* type F in gastric liquid.	[69]
**2014**	N.S.	68/F	Diplopia, nausea, weakness	Single dose of antibiotic	N.D.	20 days	F	Type F toxin in serum; *C. baratii* type F in gastric liquid and stool	[69]
**2017**	N.S.	43/F	Nausea, lethargy, unsteadiness, dysphagia, dysarthria, respiratory failure, sluggish pupils, quadriparesis, distended abdomen, minimal bowel sounds	None identified	33 days	Died from caudiopulmonary arrest 8 months after onset	A	Type A toxin in serum; *C. botulinum* type A in stool	[66]
**2017**	N.S.	5/M	Bilateral ptosis, dysarthria, mydriasis, walking difficulty, followed by constipation, muscle hypotonia, facial weakness, dilated and underactive pupils, otherwise alert	Intravenous antibiotics	Between 23 to 40 days	Discharged after 90 days	A	*C. botulinum* isolated from stool	[67]
**2017**	N.S.	33/F	Lower extremity weakness, shortness of breath, blurred vision, slurred speech, symmetric flaccid paralysis that rapidly progressed to respiratory failure requiring intubation	Previous gastric bypass surgery	N.D.	Discharged after 53 days	F	“test results positive for botulinum toxin type F”	[42]
**2018**	2017	Elderly/M	Dysarthria, dysphagia, dyspnea, ptosis, extraocular palsy, quadriparesis, intubation for respiratory failure	None identified	N.D.	Died from ventilator-associated pneumonia on day 109	A	Botulinum toxin type A in serum and stool	[82]
**2018**	N.S.	66/F	Initially lower back pain, difficulty raising arms and walking up stairs, “thick tongue”, progressive dysphagia and dysarthria, bloating, abdominal pain, and constipation; followed by worsening tachypnea and dysarthria, bilateral ptosis, descending symmetrical flaccid paralysis, complete ophthalmoplegia, ptosis, dilated pupils, absent gag reflex, dysphagia, dysarthria, inability to lift her head, and an intact sensory nervous system	Chronically immunosuppressed with oral corticosteroids; short bowel syndrome following complications of a cholecystectomy leading to ileal resection	N.D.	Discharged after 16 days; ultimately died of deep venous thrombosis, a complication of prolonged illness.	A	Botulinum toxin type A in serum and stool	[39]
**2018**	N.S.	27/M	Constipation, abdominal pain, blurry vision, bilateral ptosis, slurred speech, dysphagia, opthalmoplegia, unreactive pupils	Cyclosporine, recipient of allogeneic hematopoietic stem cell transplant	3 weeks after botulism symptom onset	Died 42 days after botulism symptom onset	A	Stool	[47]

^1^ The time given for length of colonization is the minimum. ^2^ Not determined. ^3^ Not specified. ^4^ The finding of a toxic food indicates this may be a foodborne botulism case. The authors suggest adult intestinal toxemia botulism because the patient had received previous antibiotic therapy and other individuals in the household consumed the same food but did not show symptoms.

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
