# Peer review of "Adult Intestinal Toxemia Botulism"

_toxins, 2020, doi:10.3390/toxins12020081_

Round 1

Reviewer 1 Report

It is a well-written nice short review for the topic covered and it could help readers in the BoNT related field and some medical practitioner. It should be published timely.

Some minor revision/typo correction will make the review better, such as in the middle of line 148, after the world intestine "were" should be "Where", and in line 150, "but not in always serum" should be "but not always in serum", etc. 

Author Response

We thank the reviewer for their comments on the manuscript. We have made the changes suggested in lines 148 and 150, and have checked for any other typos.

Reviewer 2 Report

This is a well organized and written overview of a rare but significant syndrome.

Minor Comments:

Lines 87-90 - Add reference to honey related botulism cases

Paragraph beginning with Line 184 - Add statement about Reference lab (i.e. State PHL or CDC) testing, as C botulinum culture and toxin testing is not typically performed by hospital labs

Author Response

The authors wish to thank the reviewer for their comments and suggestions.

A reference to infant botulism cases caused by honey consumption was added:

"The source of spores is rarely determined in adult intestinal toxemia botulism cases. This is similar to infant botulism where, with the exception of honey [26], the source of spores usually remains undiscovered."

A sentence regarding testing being limited to provincial, state, or national reference laboratories has been added:

"The task depends upon communication between clinicians, epidemiologists and provincial, state, or national reference laboratories as detection of BoNT’s or BoNT-producing clostridia is not performed by hospital laboratories."